# Numerical and Experimental Study on Melt Treatment for Large-Volume 7075 Alloy by a Modified Annular Electromagnetic Stirring

**DOI:** 10.3390/ma12050820

**Published:** 2019-03-11

**Authors:** Min He, Zhifeng Zhang, Weimin Mao, Bao Li, Yuelong Bai, Jun Xu

**Affiliations:** 1General Research Institute for Non-ferrous Metals, Beijing 100088, China; hemin5209@126.com (M.H.); bai_yuelong@163.com (Y.B.); xujun@grinm.com (J.X.); 2School of Materials and Science and Engineering, University of Science and Technology Beijing, Beijing 100083, China; weiminmao@263.net

**Keywords:** numerical simulation, electromagnetic stirring, electromagnetic field, thermal field, flow field, composition segregation, microstructure, melt treatment, grain refinement

## Abstract

This study presents a modified annular electromagnetic stirring (M-AEMS) melt treatment suitable for a large-volume and high-alloyed aluminum alloy. A 3D computational model coupling an electromagnetic model with a macroscopic heat and fluid-flow model was established by using Ansoft Maxwell 3D and Fluent from ANSYS workbench, and the effects of the electromagnetic shielding ring, the height of the magnet yoke, the shape of the iron core, and the internal cooling mandrel on the electromagnetic, thermal and flow fields were studied numerically. Based on the optimal technical parameters, the effectivity of the M-AEMS process by using 7075 alloy was validated experimentally. The results show that a favorable electromagnetic field distribution can be achieved by changing the magnet yoke height, the iron-core shape and the electromagnetic shielding ring, and the melt temperature of the 7075 alloy can drop rapidly to the pouring temperature by imposing the internal cooling mandrel; compared with ordinary annular electromagnetic stirring, the M-AEMS process creates a lower magnetic strength near the melt top, beneficial for stabilizing the melt surface; meanwhile, it yields a higher magnetic strength near the melt bottom, which increases the shear rate and ensures an optimal stirring effect. Therefore, M-AEMS works more efficiently because the thermal and composition fields become uniform in a shorter time, which reduces the average grain size and the composition segregation, and a more stable melt surface can be obtained during treatment, which reduces the number of air and oxide inclusions in the melt.

## 1. Introduction

The properties of metals have a great relationship with the solidification structure and composition distribution. Therefore, the research and development on a melt treatment method that controls the metal solidification process to improve the structure and composition distribution is a hot topic. Extensive scientific work on melt treatment was conducted. Fan developed a rheo-diecasting (RDC) process to manufacture near-net-shape components indicating almost no porosity, and a fine and uniform microstructure throughout the sample in the as-cast condition [1,2], and the continuous rheoconversion process (CRP) developed by Worcester Polytechnic Institute (WPI) [3] and the swirled enthalpy-equilibration device (SEED) process developed by Alcan [4] were applied for slurry or feedstock preparation. Recently the ultrasonic melt treatment for aluminum alloy have been increasingly studied [5,6,7]. Although these methods appear to be promising for future industrial applications, some limitations need to be overcome for mass production, such as a low productivity and shear rate, a high production cost, and a non-uniformity of microstructures.

Electromagnetic stirring, which is characterized by non-pollution, low cost, and high controllability, has been a main melt treatment method for producing Al-alloy slurry or semisolid feedstock materials in industry. The American AEMP Company first used electromagnetic stirring to produce aluminum alloy semi-solid billets from 75 to 150 mm in diameter [8]. The CREM (Casting, Refining, Electromagnetic) process [9] was proposed by Vives (33 rue Louis Pasteur, Avignon, France), and the Pechiney Company of Paris, France obtained the patent for the preparation of semi-solid metal paste by alternating current (AC) induction traveling wave electromagnetic stirring, which is used to produce a semi-solid billet of A356, A357 and AlSi6Cu3Mg aluminum alloy with a diameter of 76.2–152.4 mm [10]. Bubler in Uzwil, Switzerland, Hertwich in Braunau am Inn, Austria and EFU in Germany each developed an electromagnetic stirring horizontal continuous casting system for the production of semi-solid billets [11,12]. The Northeastern University of China developed low-frequency electromagnetic casting (LFEC) process for the production of semi-solid billets [13]. Kang studied the morphology of the primary Al phase in an A356 alloy using a self-designed horizontal electromagnetic stirrer to determine the rheological behaviors [14]. However, due to the skin effect of the alternating electromagnetic field, there exist low shear strength and poor stirring uniformity in the ordinary electromagnetic stirring process. To solve the problem, an annular electromagnetic stirring (AEMS) process was developed by GRINM (General Research Institute for Non-ferrous Metals, Beijing, China), and a computational model that couples an electromagnetic model with a macroscopic heat and fluid-flow model in a semi-solid aluminum-alloy slurry was established by using ANSYS Magnetic-Nodal (ANSYS, Inc., Pittsburgh, PA, USA) and FLOTRAN CFD (ANSYS, Inc.) programs, and its effects on microstructure and composition segregation of A357, A319 alloys semisolid billet were studied [15]. However, inhomogeneous microstructure and serious segregation unavoidably occur in large-volume melt, especially in the high-alloyed 7xxx series aluminum alloys because of the low shear rate and unsuitable flow field [16,17], and this problem cannot be solved by the current models. Moreover, most of these studies mainly focused on the DC casting billet or slab, and few research works have been done on electromagnetic stirring for complicated castings, which need more uniform temperature and composition field distribution, and shorter melt treatment time. Therefore, it is of great significance to develop large-volume, high-alloyed aluminum alloy melt treatment method.

The aim of this study is to develop a modified annular electromagnetic-stirring (M-AEMS) model, where the internal cooling module, electromagnetic shielding module and optimized electromagnetic generation module are included in the previous AEMS model to improve the cooling rate and shear rate of the system while maintaining the stability of the liquid surface. A 7075 alloy was chosen, and the effects of the cooling module, electromagnetic shielding module and optimized electromagnetic generation module on the electromagnetic field, thermal field, and flow field were investigated numerically. The effectivity of the process was experimentally verified by examining the microstructure and composition distribution of the treated castings.

## 2. Methods

### 2.1. Physical Model

Two-dimensional and three-dimensional axisymmetric physical models of AEMS and M-AEMS processes are shown in Figure 1. Compared with the AEMS, the M-AEMS is provided with other changes. The installation position of the magnet yoke and the iron core around the crucible is descended to generate a uniform rotating electromagnetic field, and an electromagnetic shielding ring is added to weaken the electromagnetic flux density near the melt surface, and cooling water is used to circulate through the internal cooling mandrel to increase the cooling rate inside the melt. Therefore, the electromagnetic field distribution of the melt in the crucible can be modified by adjusting the electromagnetic shielding ring, the magnet yoke position, the iron core structure, and the temperature gradient in the melt can be decreased via the internal cooling mandrel with changing cooling water flux, so that a stable shape of melt surface, a high shear strength at the bottom of the melt, and a more efficient melt treatment can be achieved in the M-AEMS process. 

### 2.2. Mathematical Model and Physical Properties

According to Faraday’s electromagnetic-induction law, an alloy melt is equivalent to a conductor, and when the alloy melt is placed in a rotating magnetic field, and intersects the magnetic force lines, an inductive current is generated, and the alloy melt is affected by the Lorentz force to achieve stirring.

The electromagnetic stirring process is complex because it involves an electromagnetic field, a thermal field, and a flow field, and its analysis model includes a mathematical model and a finite-element model.

In order to simulate the phenomena in the AEMS and M-AEMS processes, the following basic assumptions are made for the simulation:The melt of the 7075 alloy is an incompressible fluid;All materials are isotropic in the model;The displacement current and joule heat are ignorable.

The magnetic field is described by the Maxwell’s equations [18]:(1)$$\nabla \times \vec{E}=-\frac{\partial \vec{B}}{\partial t}$$
(2)$$\nabla \times \vec{H}=\vec{J}$$
(3)$$\nabla \cdot \vec{B}=0$$
where $\vec{B}$ is the magnetic flux density vector, $\vec{E}$ is the electric intensity vector, *t* is the time, $\vec{H}$ is the induction magnetic field intensity vector, and $\vec{J}$ is the current density vector. 

In general, Ohm’s law that defines the current density is given by:(4)$$\vec{J}=\sigma \vec{E}$$
where σ is the electrical conductivity of the melt.

The Magnetohydronamics (MHD) coupling is achieved by introducing additional source terms to the fluid momentum equation. For the fluid momentum equation, the additional source term is the Lorentz force given by:(5)$$\vec{F}=\vec{J}\times \vec{B}$$

The flow field and thermal field are governed by the equation continuity:(6)$$\nabla \cdot \vec{U}=0$$
By the momentum-conservation equation:(7)$$\frac{\partial \left(\vec{U}\right)}{\partial t}+\left(\nabla \cdot \vec{U}\right)\vec{U}=\left(\mu +\mu_{t}\right)\nabla^{2}\cdot \vec{U}-\frac{1}{\rho }\nabla \cdot p+\vec{F}$$
where $\vec{U}$ is the fluid velocity field, μ is the dynamic viscosity, μ_t_ is the turbulent viscosity, ρ is the density of the melt, and *p* is the hydrostatic pressure.

In the present study, the turbulence model is adopted. The turbulent viscosity μ_t_ is computed by combining the turbulence kinetic energy *k* and its rate of dissipation ε as follows:(8)$$\mu_{t}=\rho C_{\mu }\frac{k^{2}}{\varepsilon }$$
(9)$$\frac{\partial \left(\rho k\right)}{\partial t}+\frac{\partial }{\partial x_{i}}\left(\rho ku_{i}\right)=\frac{\partial }{\partial x_{j}}\left[\left(\mu +\frac{\mu_{t}}{\sigma_{k}}\right)\frac{\partial k}{\partial x_{i}}\right]+G_{k}-\rho \varepsilon$$
(10)$$\frac{\partial \left(\rho \varepsilon \right)}{\partial t}+\frac{\partial }{\partial x_{i}}\left(\rho \varepsilon u_{i}\right)=\frac{\partial }{\partial x_{i}}\left[\left(\mu +\frac{\mu_{t}}{\sigma_{\varepsilon }}\right)\frac{\partial \varepsilon }{\partial x_{i}}\right]+C_{1\varepsilon }\frac{\varepsilon }{k}G_{k}-C_{2\varepsilon }\rho \frac{\varepsilon^{2}}{k}$$
where $G_{k}$ represents the generation of turbulence kinetic energy due to the mean velocity gradients, $u_{i}$ is the velocity component, $C_{1\varepsilon }$, $C_{2\varepsilon }$ and $C_{\mu }$ are constants, and the values are 1.44, 1.92 and 0.09, respectively. $\sigma_{k}$ and $\sigma_{\varepsilon }$ are the turbulent Prandtl numbers for *k* and ε, the values are 1.0 and 1.3, respectively.

Thermal field are governed by the energy-conservation equation:(11)$$\frac{\partial \left(\rho T\right)}{\partial t}+\nabla \cdot \left(\rho \vec{U}T\right)=\nabla \cdot \left(\frac{k}{c_{p}}\nabla T\right)$$
where *c_p_* is the heat capacity at constant pressure, *k* is thermal conductivity, and *T* is the temperature.

The temperature-dependent physical parameters of the 7075 alloy, i.e., density, electrical resistivity, specific heat, thermal conductivity, and liquid viscosity, are calculated by JMatPro 7.0 software (Sente Software Ltd., Guildford, UK), as illustrated in Figure 2. These physical parameters are very important for simulating the flow field and thermal field of the 7075 alloy melt.

### 2.3. Finite-Element Model and Boundary Conditions

The simulation process is performed by following steps:A physical model was established using the UG 9.0 software (Siemens PLM Software, Plano, TX, USA).The physical model was imported into Ansoft Maxwell (15.0, ANSYS, Inc.) to obtain the magnetic-flux density B by calculating the electromagnetic field.Through the MHD module, B is imported as the source term into Ansoft Fluent (15.0, ANSYS, Inc.).The temperature and flow fields are meshed and simulated numerically by using Ansoft Fluent.

In this case, the intensity of the melt flow can be using the non-dimensional magnetic Taylor number *Ta_m_* [19],
(12)$$Ta_{m}=\frac{B^{2}R^{4}\sigma \pi f}{\rho v^{2}}$$
where *B* denotes the amplitude of the magnetic field, *R* the crucible radius, σ the electrical conductivity of the melt, *f* the rotational frequency of the magnet, ρ the density of the melt and *ν* the kinematic viscosity. Inserting the respective parameters and the material properties of 7075 alloy into Equation (12) results in magnetic Taylor numbers of *Ta_m_* = 1.86 × 10^5^ and *Ta_m_* = 2.05 × 10^5^ for AEMS and M-AEMS, respectively. The critical value for a transition from a laminar to a turbulent flow in a finite circular cylinder of aspect ratio is *Ta_m_,cr* = 1.23 × 10^5^ [20]. Above the critical value of the magnetic Taylor number *Ta_m_,cr*, the rotating electromagnetic field driven flow turns to turbulence. Typically, the turbulence model (*k*, e-model) was adopted in the numerical simulation of AEMS and M-AEMS processes.

A schematic view of axisymmetric calculation domain and its mesh plot in the heat-transfer model is shown in Figure 3, and the detailed thermal boundary conditions are listed in Table 1. Numerical calculation was performed with a commercial finite-element software ANSYS Workbench 15.0 (ANSYS, Inc., Pittsburgh, PA, USA). The finite-element model consists of 178,092 uniform hexahedral elements.

### 2.4. Experimental

A commercial 7075 alloy was used in the verification test, and its chemical composition is listed in the Table 2. The differential-scanning calorimetry curves of the 7075 alloy are shown in Figure 4, and the solidus and liquidus temperatures are 741 and 908 K, respectively. Determining the solidus liquidus temperature of the 7075 alloy is useful for precisely controlling the temperature of the melt during the experiment. The experimental procedure was carried on as followed. The 7075 alloy was smelted at 1050 K in a resistance-heated furnace (FNS Electric Furnace Co., Ltd., Beijing, China). After degassing and holding for 10 min, the melt was cooled to 970 K, and poured into a stirring crucible preheated at 500 K in advance, at the same time, the electromagnetic stirring device was initiated, and the temperature of the melt was measured. The mass of the treated melt was about 40 kg in each test. When the melt in the stirring crucible was cooled to 908 K, it was poured into an iron-cast mold with an inner diameter of 300 mm and depth of 200 mm, and cooled in air to ambient temperature. The magnetic-flux density was measured using a gauss meter (Hengtong Magnetoelectricity Co., Ltd., Shanghai, China). Time–temperature history data were collected by two K-type thermocouples and a temperature collector (TOPRIE Electronic Co., Ltd., Shenzhen, China). The positions of the measured points for the melt temperature and magnetic-flux density are shown in Figure 1a,c. Comparative trials of both AEMS and M-AEMS processes were performed, and the casting experimental process and the electromagnetic stirring parameters in two cases were the same except for different electromagnetic stirrer structures. In this study, the frequency of the stirring current is 10 Hz, and the intensity of stirring current is 60 A.

In order to catch the melt surface shape, a pure foil with the same width as the annulus was inserted into the alloy melt during electromagnetic stirring. The portion of the aluminum foil immersed in the melt was melted, and the contour of the melt liquid surface shape was retained [21]. For the metallographic inspection, disk pieces were first taken in the positions of the half thickness of the casting samples, and the composition measurement points and the metallographic sampling positions for the disks piece are shown in Figure 5. The points of M_1_, M_2_ and M_3_ denote metallographic sampling positions, which are taken at 5, 75 and 145 mm from the center of the disk piece, respectively. The points from C_1_ to C_9_ are sampling positions for composition measurement, respectively, which are taken at 5, 22.5, 40, 57.5, 75, 92.5, 110, 127.5 and 145 mm from the center of the disk piece. The metallographic samples were anodized with Barker’s reagent (4% HBF4 in distilled water) and examined under polarized light using the optical microscope (OM, Zeiss Axiovert 2000MAT, Carl Zeiss AG, Heidenheim and der Brenz, Germany). The grain size was measured using the linear intercept method (ASTM E112-10 [22]). The chemical composition of the cross section of the disk was evaluated by means of the Optical Emission Spectrometer (Foundry-Master Pro, Oxford Instruments, Taunusstein, Germany).

## 3. Results

### 3.1. Electromagnetic Field Analysis

The electromagnetic field distribution in the longitudinal section of the 7075 alloy melt in the stirring crucible by AEMS and M-AEMS obtained by numerical simulation are shown in Figure 6. It is noted that, in two cases, the largest magnetic flux density lies near the wall of the stirring crucible, and the magnetic flux density decreases gradually from the wall of the stirring crucible to the wall of the central cooling mandrel because of the “skin effect”. As shown in Figure 6a, in the case of the AEMS, the distribution of the magnetic flux density at the melt top surface is similar to that at the bottom of the melt, the smallest magnetic flux density exists at both ends of the melt, and the largest magnetic field occurs in the center part of the stirring crucible wall. In the case of M-AEMS, the magnetic flux density becomes small at the melt top surface, and the largest magnetic flux field lies near the bottom of the stirring crucible, as shown in Figure 6b.

The magnetic flux density data collected along a dashed line near the crucible wall in cases of AEMS and M-AEMS are shown in Figure 7. In comparison to the two curves, the magnetic flux density at the melt top surface decreases from 8.2 mT in the AEMS to 3.7 mT in the M-AEMS, but the magnetic fluid density at the bottom of the melt increases from 8.3 mT in the AEMS to 21.2 mT in the M-AEMS.

### 3.2. Thermal Field and Flow Field Analysis

The velocity field distributions in the cross section of the 7075 alloy melt in the stirring crucible by AEMS and M-AEMS are shown in Figure 8. It is clearly seen that, the velocity field distribution on the melt top surface is similar to that on the melt bottom, as shown in Figure 8a,b, and the maximum velocity is 1.51 m·s^−1^. In the case of M-AEMS, the velocity on the melt top surface is very small, and the maximum velocity in the flow field is 0.28 m·s^−1^, as shown in Figure 8c; but the velocity at the melt bottom becomes very large, and the maximum velocity in the flow field is 14.31 m·s^−1^, as shown in Figure 8d. The flow field distribution is almost consistent with the electromagnetic field distribution.

Figure 9 shows the numerically calculated temperature-time curves of the 7075 melt in the stirring crucible at thermocouple-1 (T_1_) and thermocouple-2 (T_2_) in cases of AEMS and M-AEMS. The position of T_1_ and T_2_ are shown in Figure 1. For the AEMS, the difference between temperatures of T_1_ and T_2_ is large initially. The large difference decreases slightly with time. After 15 s of processing, temperature of T_1_ and T_2_ are 910 and 882 K, respectively. Because temperature of T_2_ is already below the liquidus temperature of the 7075 alloy, it does not meet application requirements. For the M-AEMS, the difference between temperature of T_1_ and T_2_ is much smaller than that of the AEMS initially, and the difference becomes very small after 8 s. After 15 s, the temperatures of T_1_ and T_2_ are 909 and 907.5 K, respectively. Therefore, the M-AEMS can significantly improve the uniformity of the thermal field of the 7075 alloy melt in a short time.

### 3.3. Experimental Validation

The magnetic-flux density distributions of the 7075 alloy melt without charge along the stirring crucible wall obtained numerically and experimentally in the M-AEMS are shown in Figure 10. It is noted that the magnetic flux density increases initially, and then decreases with an increase in melt height. The numerical magnetic flux density results agree well with the experimental ones.

Time-temperature histories of T_1_ and T_2_ obtained by simulation and experiment are shown in Figure 11. It is clearly seen that the numerical and experimental temperatures decrease with stirring time. Although the rate of the temperature reduction in the experiment is lower than that in the simulation, the change rule and trend are similar. Both at T_1_ and T_2_, in the numerical temperature curve, 15 s are taken to drop to the pouring temperature, while the experimental curve took about 21 s.

The Zn, Mg and Cu elements distributions in the radial direction of the disk piece with pouring temperatures of 910, 920, 930 and 940 K respectively are shown in Figure 12. Compared with the AEMS, segregations of Zn, Mg and Cu elements are improved greatly in the M-AEMS. As shown in Figure 12a, on the condition with a pouring temperature of 910 K, the average segregation rates of the Zn, Mg and Cu elements at all measuring points on the disk piece were remarkably reduced from 4.4%, 5.9% and 6.5%, to 2.1%, 3.3% and 4.2%, respectively, and also exists the same effect to alleviate composition segregation when the pouring temperatures are 920, 930 and 940 K, respectively, as shown in Figure 12b.

The microstructure of the disk piece treated by AEMS and M-AEMS at a pouring temperature of 910 K is shown in Figure 13. Compared with the AEMS, the microstructure of the disk piece treated by M-AEMS is significantly finer. The statistics data of average grain size of the disk piece treated by AEMS and M-AEMS with the pouring temperatures of 910, 920, 930 and 940 K at different sampling locations are shown in Figure 14. The average grain size is counted by a large number of samples. As shown in Figure 14, similar to 910 K, when the pouring temperature is 920, 930 and 940 K, the effect of M-AEMS on grain refinement still exists.

## 4. Discussion

By designing the new electromagnetic generator structure, the electromagnetic field distribution can be changed purposefully. Compared with the AEMS, the magnetic flux density of the M-AEMS is lower near the top of the melt, and significantly higher near the melt bottom, where the magnetic field is also more homogeneous.

Quite a different flow field is attained due to the change of the electromagnetic field distribution. Meanwhile, the maximum velocity increased from 1.51 m·s^−1^ in the AEMS to 14.31 m·s^−1^ in the M-AEMS with a stable liquid surface obtained. Figure 15 shows the shapes of the aluminum foil melted in the 7075 alloy melt with and without an electromagnetic shielding ring, which can reflect the shape of the melt surface vividly. The shape of the aluminum foil with the electromagnetic shielding ring is more gradual, and this stable surface of the 7075 alloy melt is beneficial to reduce the entrainment of the air and oxidation inclusions.

The water-cooling mandrel increases the cooling capacity of the M-AEMS, and the optimized electromagnetic field increases the shear rate in the meantime. High shear rate promotes heat exchange and mass transfer between high temperature and low temperature flows. Under the joint action, more uniform thermal field and composition field are obtained in the melt of 7075 alloy, and a fixed pouring temperature can be reached in a short time. 

For the present model, numerical magnetic flux density results agree well with the experimental ones, but there exists a gap between the numerical and experimental temperature results, where 15 s are needed to reach the pouring temperature according to the numerical results, while 21 s are measured in the experiment. There are possible reasons leading to the temperature difference between simulations and experiments. Firstly, the accuracy of the thermodynamic boundary conditions and the physical parameters of the materials used in numerical simulations, such as convective heat transfer between stirring crucible and air, heat exchange between the cooling mandrel and the melt, and the viscosity of the 7075 alloy melt, etc. are also responsible for the gap between the numerical and the experimental results. Secondly, the electromagnetic field was simulated by Maxwell 3D, and the flow field and thermal field were simulated by Fluent software in ANSYS Workbench. There are differences between the meshes of Maxwell and Fluent, which may be a reason for the gap between simulations and experiments.

In general, a lower magnetic field strength near the melt top stabilizes the liquid surface, and a higher magnetic field strength near the melt bottom can generate a higher shear rate and a stronger stirring effect. Therefore, a more stable liquid surface and a stronger stirring effect are expected for the applied M-AEMS. A more stable liquid surface is beneficial to reduce the air entrainment and oxidation inclusion in the melt. A strong stirring rate can improve the uniformity of thermal field and composition field for 7075 alloy melt. It is well known that the condition that equiaxed grains appear ahead of the columnar front can be expressed as [23]: (13)$$G<0.617N_{0}^{\frac{1}{3}}\left[1-{\left(\frac{\Delta T_{N}}{T_{C}}\right)}^{3}\right]\Delta T_{C}$$
where *G* is the temperature gradient in the melt ahead of solid–liquid interface; *N*_0_ is the heterogeneous nucleation rate; Δ*T_N_* is the critical undercooling of heterogeneous nucleation and Δ*T_C_* is the undercooling in the melt ahead of the solid–liquid interface. 

It can be seen from Equation (13) that reducing *G* and increasing *N*_0_, Δ*T_C_* can promote transferring coarse columnar grains to fine equiaxed grains, which is favorable to the equiaxed fraction increment and the grain refinement. In the presence of optimal AEMS, the more intense forced convection generated by the electromagnetic stirring can cause the dendrite fragmentation and the dendrite melting broken at the wall of stirring crucible and cooling mandrel, which increases the heterogeneous nucleation rate *N*_0_. Furthermore, the forced convection in the stirring crucible can also accelerate the uniform of the bath solute and temperature, which increases the undercooling Δ*T_C_* and reduces the temperature gradient *G* [24]. Therefore, such a melt has numerous heterogeneous nucleation particles and a large degree of undercooling, and a small temperature gradient can promote grain refinement and uniform composition of the casting. Therefore, the segregation of components in the casting is reduced and the grain size is refined and uniform.

## 5. Conclusions

A numerical and experimental study on M-AEMS melt treatment for a large-volume 7075 alloy was conducted. A 3D computational model coupling with an electromagnetic model with a macroscopic heat and fluid-flow model was established, and the effects of the electromagnetic stirrer configuration parameters on the electromagnetic, thermal and flow fields were studied numerically. Based on the optimal results, the effectivity of the M-AEMS process was validated experimentally, and the conclusions are made as follows:Compared with the ordinary AEMS process, electromagnetic shielding module reduces the magnetic flux density near the melt top, which ensures a more stable liquid surface; optimized electromagnetic generation module creates a higher magnetic flux density near the melt bottom, which exerts a higher shear rate and stronger stirring effects. The magnetic flux density near the melt top decreases from 8.2 to 3.7 mT, and the magnetic flux density near the melt bottom increases from 8.3 to 21.1 mT. The maximum velocity of the flow field increased from 1.51 to 14.31 m·s^−1^ for the M-AEMS. A uniform thermal field and a stable liquid surface can be achieved in a shorter time, and the temperature difference between the crucible wall and mandrel wall is 1.5 K after 21 s of melt treatment when the temperature of 7075 alloy melt drops from 970 to 907.5 K.By the M-AEMS process, the microstructure of the disk castings becomes more uniform and finer. The average grain size in the radial direction of the casting with pouring temperature of 910, 920, 930 and 940 K was reduced from 127, 144, 171 and 187 microns, respectively, to 115, 131, 152 and 174 microns. In addition, the chemical composition is distributed homogeneously, the relative segregation degree of the main alloying elements of Zn, Mg and Cu were reduced from 4.4%, 5.9% and 7.5%, respectively, to 2.1%, 3.3% and 4.2%. Both finer microstructure and more uniform composition distribution of the castings were obtained at the pouring temperatures of 910, 920, 930 and 940 K, indicating that the M-AEMS process has a wide process window to pouring temperature.The parameters of thermodynamic boundary conditions and indirectly coupled calculation have a great influence on the gap between numerical and experimental results. Thus, further research work should be done to realize directly coupled calculation of the electromagnetic field, thermal field and flow field for obtaining a precise prediction.

## Figures and Tables

**Figure 1 materials-12-00820-f001:**
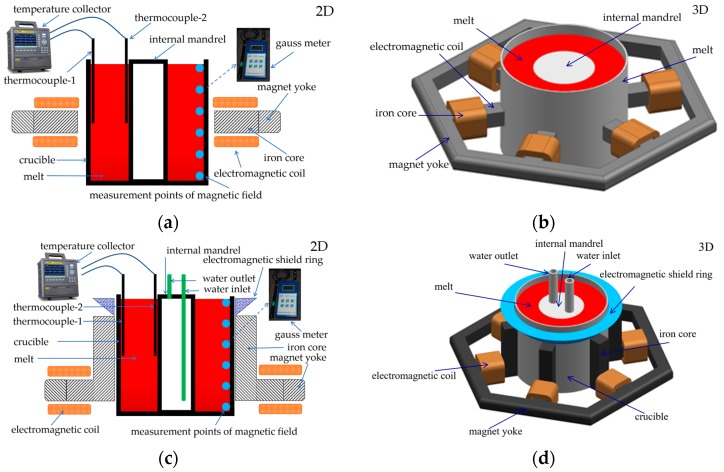
Two-dimensional and three-dimensional geometry models of AEMS and M-AEMS created by using UG: (**a**) 2D of AEMS; (**b**) 3D of AEMS; (**c**) 2D of M-AEMS; (**d**) 3D of M-AEMS.

**Figure 2 materials-12-00820-f002:**
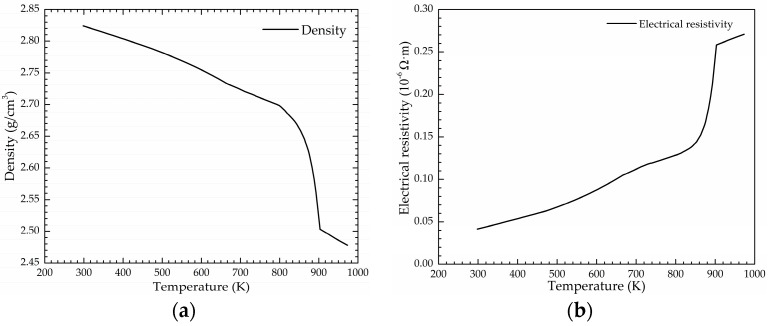
The temperature-dependent physical properties of the 7075 alloy: (**a**) density; (**b**) electrical resistivity; (**c**) specific heat; (**d**) thermal conductivity; (**e**) liquid viscosity.

**Figure 3 materials-12-00820-f003:**
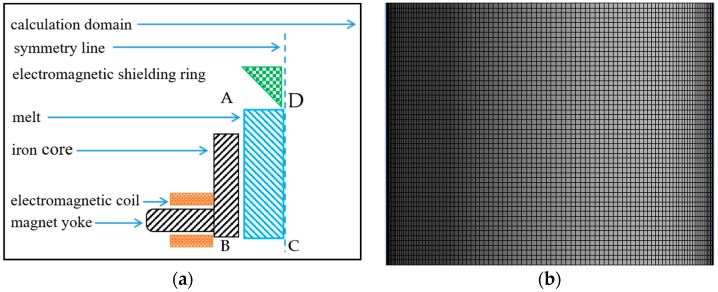
A schematic view of axisymmetric calculation domain and its mesh plot in the heat-transfer model: (**a**) axisymmetric calculation domains and (**b**) mesh plot.

**Figure 4 materials-12-00820-f004:**
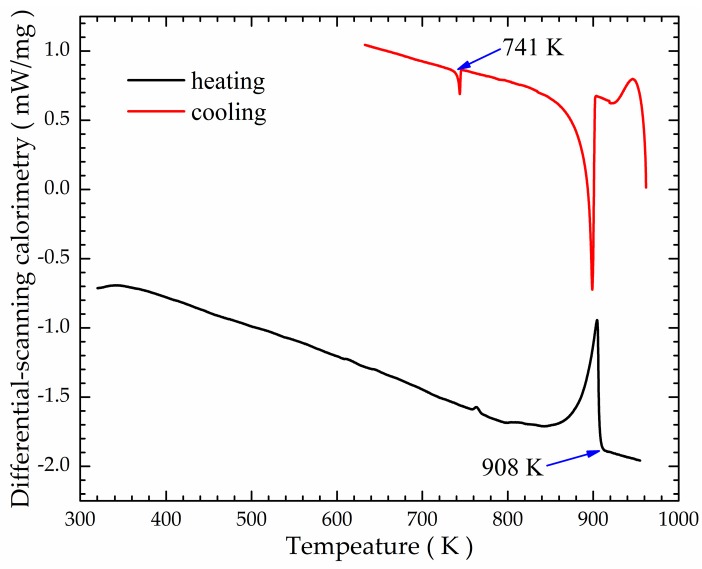
Differential-scanning-calorimetry curves of 7075 alloy.

**Figure 5 materials-12-00820-f005:**
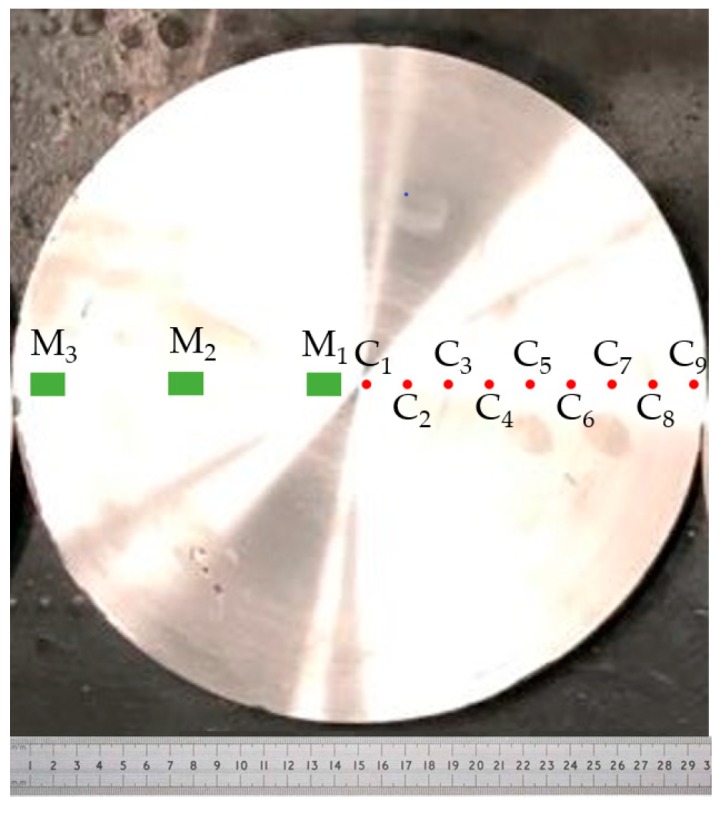
The composition measurement points from C_1_ to C_9_ and the metallographic samples positions (M_1_, M_2_ and M_3_) on the disk piece.

**Figure 6 materials-12-00820-f006:**
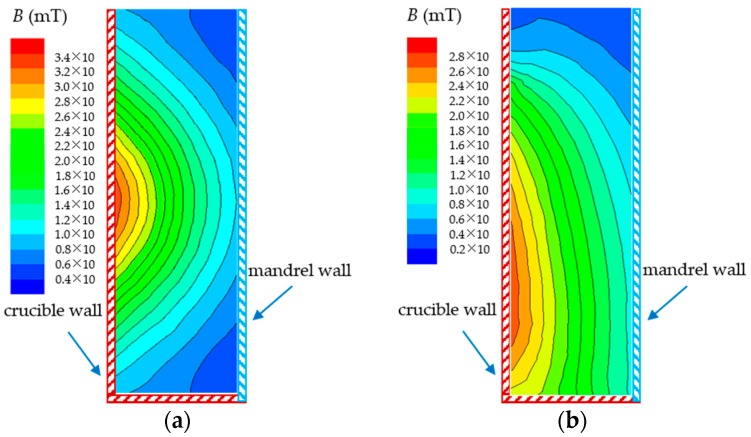
Electromagnetic field distribution in the longitudinal section of the 7075 alloy melt in a stirring crucible obtained by numerical simulation in two cases: (**a**) AEMS and (**b**) M-AEMS.

**Figure 7 materials-12-00820-f007:**
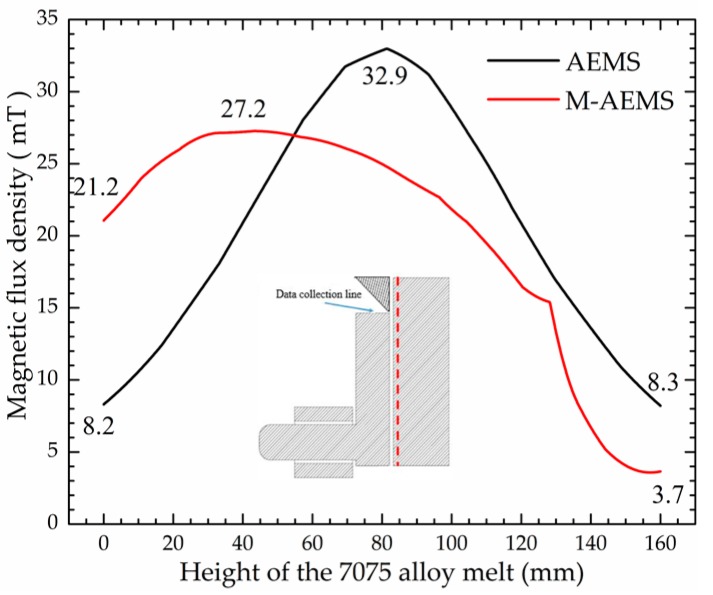
Comparison between the magnetic flux densities along a dashed line near the wall of the stirring crucible in the cases of AEMS and M-AEMS.

**Figure 8 materials-12-00820-f008:**
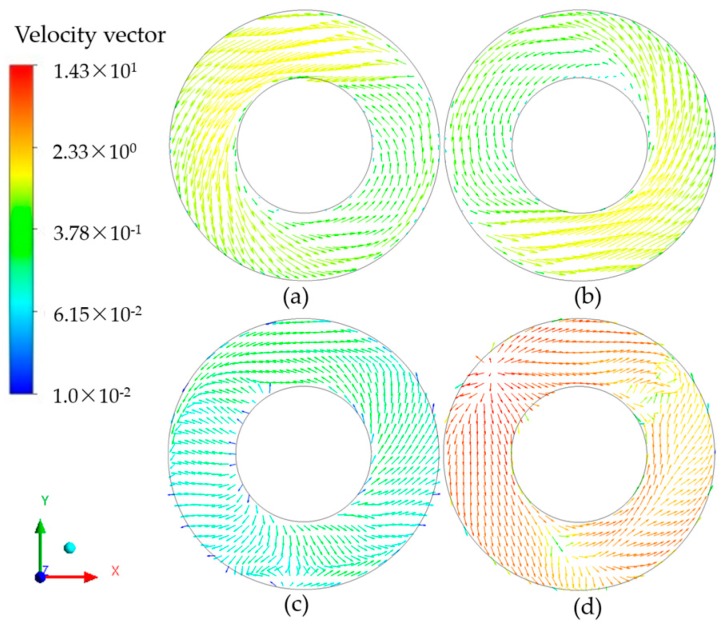
Velocity field distributions in the cross section of 7075 alloy melt in the stirring crucible by AEMS and M-AEMS: (**a**) velocity field distribution on the melt top surface of AEMS; (**b**) velocity field distribution on the melt bottom of AEMS; (**c**) velocity field distribution on the melt top surface of M-AEMS; (**d**) velocity field distribution on the melt bottom of M-AEMS.

**Figure 9 materials-12-00820-f009:**
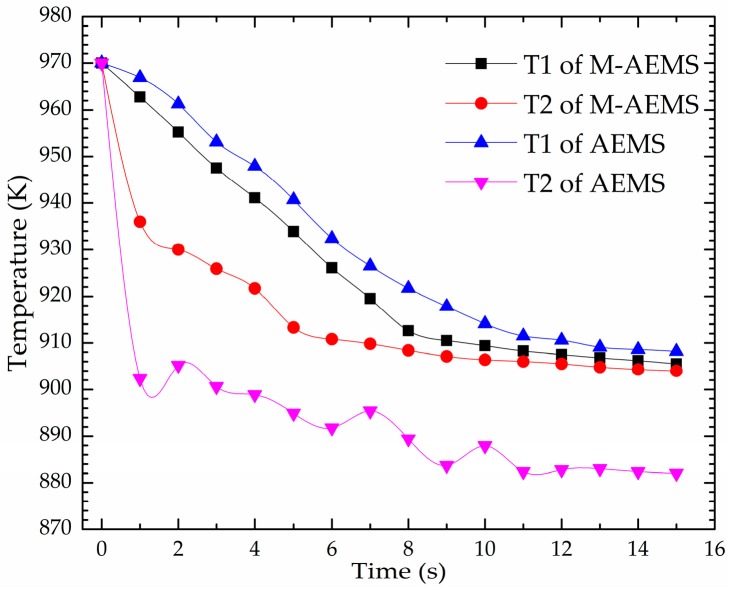
The numerically calculated temperature-time curves of the 7075 melt in the stirring crucible at the T_1_ and T_2_ points in cases of AEMS and M-AEMS.

**Figure 10 materials-12-00820-f010:**
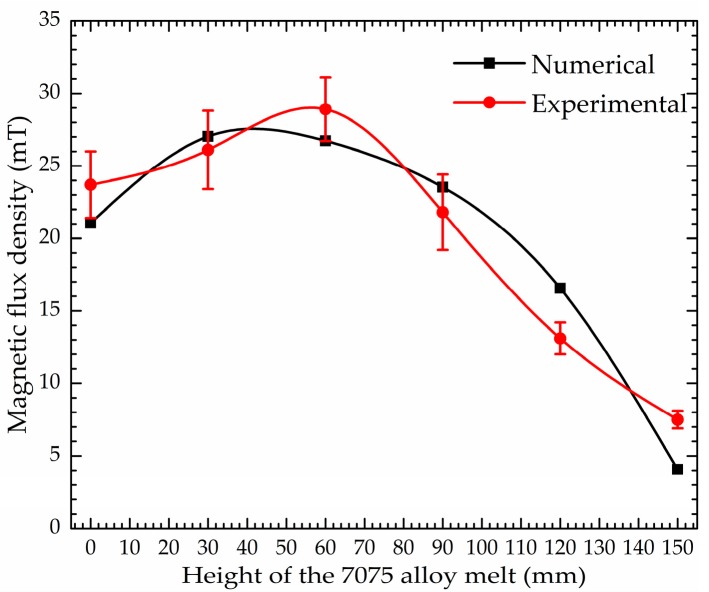
Comparison between simulated and measured magnetic flux density.

**Figure 11 materials-12-00820-f011:**
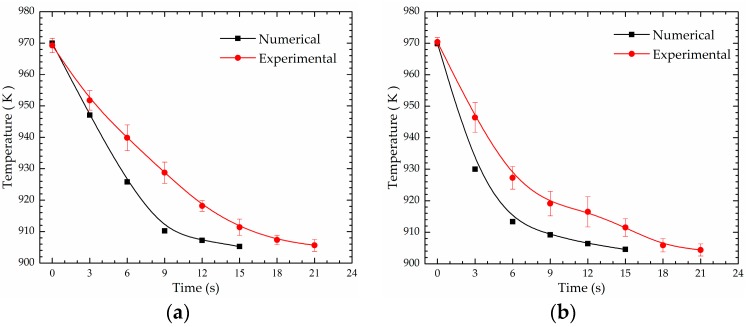
Comparison between simulated and measured time-temperature histories: (**a**) T_1_ and (**b**) T_2_.

**Figure 12 materials-12-00820-f012:**
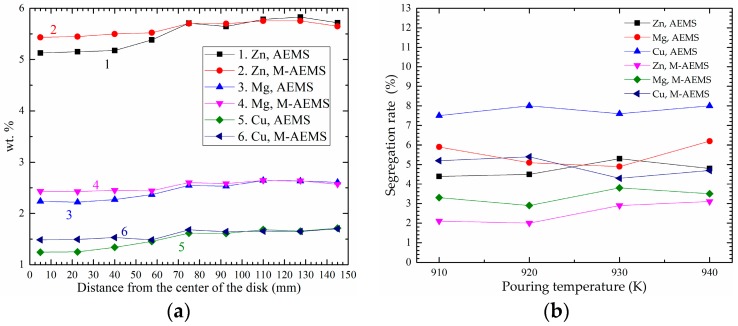
(**a**) The Zn, Mg and Cu elements distribution and (**b**) segregation rates with different pouring temperature in the radial direction of the disk pieces by AEMS and M-AEMS.

**Figure 13 materials-12-00820-f013:**
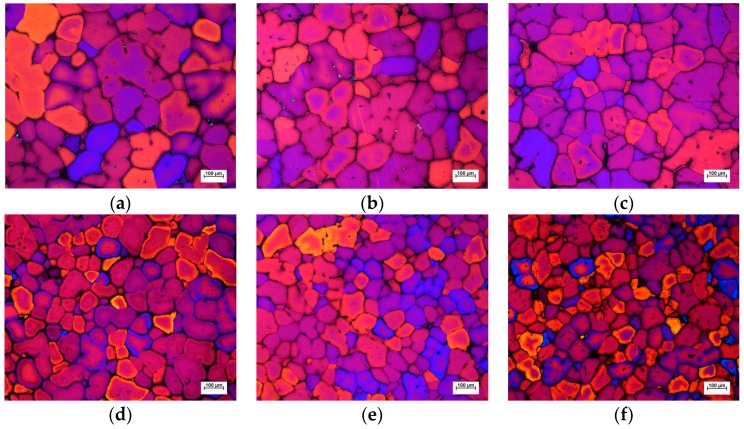
The microstructure of the disk pieces treated by AEMS and M-AEMS with the pouring temperature of 910 K at different sampling locations: (**a**) M_1_, AEMS; (**b**) M_2_, AEMS; (**c**) M_3_, AEMS; (**d**) M_1_, M-AEMS; (**e**) M_2_, M-AEMS; (**f**) M_3_, M-AEMS.

**Figure 14 materials-12-00820-f014:**
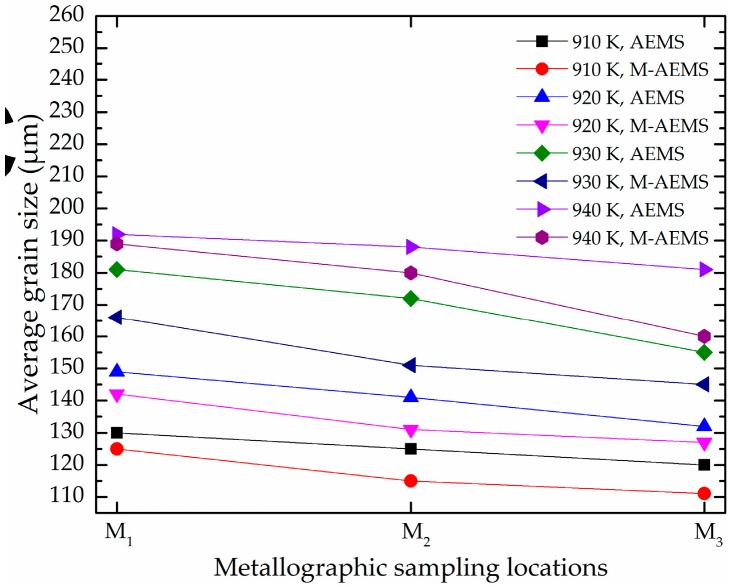
The average grain size of the disk pieces treated by AEMS and M-AEMS with the pouring temperatures of 910, 920, 930 and 940 K at different sampling locations.

**Figure 15 materials-12-00820-f015:**
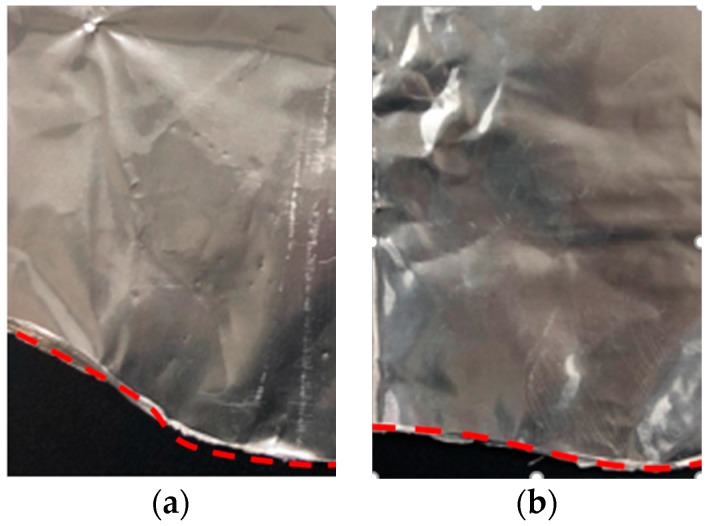
The shapes of the pure aluminum foil melted in the 7075 alloy melt: (**a**) without electromagnetic shielding ring; (**b**) with electromagnetic shielding ring.

**Table 1 materials-12-00820-t001:** Boundary conditions used in this study.

Boundary Conditions	Type		Heat Transfer Coefficient (W/m^2^·K)	Free Stream Temperature (K)	Heat Generation Rate (W/m^3^)
AB	Convection		20	300	0
BC	Adiabatic		–	–	0
CD	Convection	AEMS	20	300	0
		M-AEMS	2000	300	0
DA	Convection		20	300	0

**Table 2 materials-12-00820-t002:** Chemical composition of the 7075 alloy in this study (wt.%).

Zn	Mg	Cu	Cr	Fe	Si	Ti	Mn	Other Each Element	Al
5.63	2.41	1.57	0.15	0.19	0.36	0.14	0.27	<0.05	Balance

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
