# Peer review of "Numerical and Experimental Study on Melt Treatment for Large-Volume 7075 Alloy by a Modified Annular Electromagnetic Stirring"

_materials, 2019, doi:10.3390/ma12050820_

Reviewer 1 Report

I got acquainted with great interest with the work “Numerical and Experimental Study on Melting for Large-Volume(d?) 7075 alloy by a modified annular electromagnetic stirring”, since external influences on melts constitute the field of my scientific interests. Therefore, first of all, thanks to the authors for the interesting work.

However, I have a few questions and comments in the authors:

line 3 - in the title of the article, the authors talk about processing a large volume of alloy. But nowhere else in the text is neither the mass nor the volume of the metal being processed.

Line 41 - missing a space before referring to the literature.

Line 89-93 - in this paragraph, the authors describe the changes made to the M-AEMS system. But the reasons and logic according to which they were carried out are not voiced by the authors. For example, why “the height of the magnetic yoke was redused ..”? Why not promoted?

Line 98 - According to the above model, the melt is in the process of processing in a closed system (not flow through), but due to the lack of dimensions in the diagram, it is difficult to estimate the volume of the metal being processed. From my point of view, it was necessary to give the primary model, that the differences between the old and the new system were visible. Moreover, the experimental melting was carried out in both systems.

Line 99 - on the figure caption (a) - 2D  ,( b) - 3 D are marked. But in the figure itself the designations (a) and (b) are not used.

Equation (1)-(2) - these equations are absolutely identical. It is not clear why duplicate them. Most likely there is a mistake.

Line 160 - based on the size of the cast ingot, you can roughly determine the mass of the melt of the order of 38 kg. But it is not clear, is it the whole volume of the processed metal?

Line 166 - describes the areas for determining the chemical composition and microstructure research on a cast disk. However, it is not indicated in which part of the ingot measurements were taken. On the surface, or ½ thickness?

Line 227 - according to the distribution of the magnetic field previously shown in Figure 6, one would have expected a greater difference between the reading values of “T1 of M-AEMS” and “T1 of AEMS”. However, in Figure 10 (b) these temperatures are very close. How do the authors explain this?

Line 250 - a typo in the word "respectively."

Line 284 - a typo in the figure number - should have written 17. And in the signature of the drawing on Line 289 too.

Line 307 is the wrong order number of the equation. Instead of 13 you should write 14. and, therefore, in Line 310 too.

Line 342 - the grain size changes in the conclusions are incorrect. Since, according to Figure 16, grain size measurements were carried out for two systems and 4 casting  temperatures. In the conclusion, it is not indicated for what temperature the grain sizes are given. Moreover, there is no point on the graph at the axis of the ingot with such a temperature.

Author Response

Dear Editors and Reviewers:

        Thank you for your letter and for the reviewer’s comments concerning our manuscript entitled “Numerical and Experimental Study on Melt Treatment for Large-volume 7075 Alloy by a Modified Annular Electromagnetic Stirring” (ID: materials-449969). Those comments all are very valuable and very helpful for revising and improving our paper, as well as the important guiding significance to our study. We are pleased that you are interested in our work. We have studied comments carefully and have made correction which we hope meet with approval. Revised portion are marked in red in the paper. The revised version and the responds to the reviewer’s comments are attached.

        Once again,  special thanks to you for your quality comments and suggestion.

Reviewer 2 Report

My report is in the attached file named “Reviewer Comments”.

Author Response

Dear Editors and Reviewers:

        Thank you for your letter and for the reviewer’s comments concerning our manuscript entitled “Numerical and Experimental Study on Melt Treatment for Large-volume 7075 Alloy by a Modified Annular Electromagnetic Stirring” (ID: materials-449969). Those comments all are very valuable and very helpful for revising and improving our paper, as well as the important guiding significance to our study. We have studied comments carefully and have made correction which we hope meet with approval. Revised portion are marked in red in the paper. The revised version and the responds to the reviewer’s comments are attached.

        Once again, thanks your very much for you comments and suggestion.

Reviewer 3 Report

All the quantities from Equation(1) - (12) are to be explained (such q from 3).

Details regarding the simulations presented in the results sections would be indicated.

the quality of Figures 8 and 9 is to be increased.

English is to be improved in the results and conclusion section

Author Response

Dear Editors and Reviewers:

        Thank you for your letter and for the reviewer’s comments concerning our manuscript entitled “Numerical and Experimental Study on Melt Treatment for Large-volume 7075 Alloy by a Modified Annular Electromagnetic Stirring” (ID: materials-449969). Those comments all are very valuable and very helpful for revising and improving our paper, as well as the important guiding significance to our study. We have studied comments carefully and have made correction which we hope meet with approval. Revised portion are marked in red in the paper. The main corrections in the paper and the responds to the reviewer’s comments are attached.

        Special thanks to you for your quality comments.

Round  2

Reviewer 2 Report

Comments: 

The authors have replayed to the comments in the reviewer report. The authors have added the modifications as a function of the remarks and suggestion of the reviewer. Therefore, the quality of this new version is also improved.

In the revised version, the corrections added by the authors are in red color and the removed sentence are in blue color. The authors could present a new version of the paper include the new corrections with respect to the template and the style imposed by the Materials journal.